# Magnetite-Incorporated 1D Carbon Nanostructure Hybrids for Electromagnetic Interference Shielding

**DOI:** 10.3390/nano14151291

**Published:** 2024-07-31

**Authors:** Bayan Kaidar, Aigerim Imash, Gaukhar Smagulova, Aruzhan Keneshbekova, Ramazan Kazhdanbekov, Eleonora Yensep, Doszhan Akalim, Aidos Lesbayev

**Affiliations:** 1Department of “General Physics”, Intistute of Energy and Mechanical Engineering Named after A. Burkitbayev, Satbayev University, 22a Satpaev Str., Almaty 050013, Kazakhstan; kaydar.bayan@gmail.com (B.K.); imash.aigerim@icp.kz (A.I.); kazhdanbekov_ramazan@live.kaznu.kz (R.K.); yensep_eleonora2@live.kaznu.kz (E.Y.); d.akalim@stud.satbayev.university (D.A.); a.lesbayev@satbayev.university (A.L.); 2Institute of Combustion Problems, 172 Bogenbay Batyr Str., Almaty 050012, Kazakhstan; a.keneshbekova@icp.kz; 3Faculty of Chemistry and Chemical Technology, Al Farabi Kazakh National University, 71 al-Farabi Ave., Almaty 050040, Kazakhstan

**Keywords:** electromagnetic interference shielding, environmental protection, magnetite, carbon nanotubes, carbon fibers, hybrid composites

## Abstract

The increasing reliance on electronic technologies has elevated the urgency of effective electromagnetic interference (EMI) shielding materials. This review explores the development and potential of magnetite-incorporated one-dimensional (1D) carbon nanostructure hybrids, focusing on their unique properties and synthesis methods. By combining magnetite’s magnetic properties with the electrical conductivity and mechanical strength of carbon nanostructures such as carbon nanotubes (CNTs) and carbon fibers (CFs), these hybrids offer superior EMI shielding performance. Various synthesis techniques, including solvothermal synthesis, in situ growth, and electrostatic self-assembly, are discussed in detail, highlighting their impact on the structure and properties of the resulting composites. This review also addresses the challenges in achieving homogeneous dispersion of nanofillers and the environmental and economic considerations of large-scale production. The hybrid materials’ multifunctionality, including enhanced mechanical strength, thermal stability, and environmental resistance, underscores their suitability for advanced applications in aerospace, electronics, and environmental protection. Future research directions focus on optimizing synthesis processes and exploring new hybrid configurations to further improve electromagnetic properties and practical applicability.

## 1. Introduction and Background

The problem of protection from electromagnetic interference (EMI) is becoming increasingly urgent as society becomes more dependent on electronic technologies. The intensive development of technology and the associated increase in electromagnetic interference creates a need for new materials that can provide reliable shielding without compromising the functionality and portability of devices [1,2,3,4,5,6,7,8,9,10]. Electromagnetic interference has a significant impact on the environment, human health, and the functioning of electronic equipment, which requires careful study and analysis [11,12]. These aspects require an integrated approach, including scientific research and practical measures to reduce the negative effects of electromagnetic interference.

The electromagnetic shielding of materials can be evaluated by quantifying loss resulting from reflection, absorption, and/or multiple reflections of incoming electromagnetic waves. Traditional metal shields, while effective in reflecting and absorbing interference, have several significant drawbacks that limit their use in today’s high-tech and portable devices. Metals such as copper and aluminum, while highly conductive, have significant mass and are prone to corrosion, which requires additional protection and maintenance costs [13]. Moreover, the use of metals in shielding often leads to the creation of secondary sources of electromagnetic contamination due to strong signal reflections [14], which is particularly critical in sensitive applications such as medical devices [15] and aerospace engineering [16].

The stable growth in the number of scientific publications in the field of electromagnetic shielding since 2014, as observed in the Scopus and Web of Science databases, indicates the expansion of the boundaries of knowledge and the deepening of the understanding of the phenomenon of electromagnetic shielding (Figure 1). Recent studies have focused on the development and optimization of nanostructured composites comprising metals and their oxides, conducting polymers, carbon nanotubes, graphene, and MXene [17,18,19,20,21,22,23]. However, despite significant advances, existing research faces several challenges and gaps, such as the susceptibility of carbon-metal hybrids to oxidation, which reduces their durability and efficiency [24]. The control of the morphology and distribution of nanofillers within the polymer matrix is often challenging, resulting in heterogeneous properties that impair the predictability and reproducibility of the performance of protective materials [25].

Analyses of available materials show that despite good protective properties, many of them do not fulfill the requirements in terms of weight, flexibility, or cost-effectiveness. This has led to the investigation of hybrid materials that combine the strengths of different components and overcome the limitations of individual materials. Hybrids based on 1D/2D carbon nanostructures with the inclusion of magnetite represent a promising solution for EMI shielding due to their unique combination of properties. The inclusion of magnetite enhances the magnetic properties of carbon nanostructures, making them more effective in deflecting and absorbing electromagnetic waves. Furthermore, the structural versatility of carbon nanostructures allows for customized designs to meet specific protection requirements.

Nevertheless, important challenges such as large-scale synthesis, durability, and environmental impact of fabrication remain unresolved. This review aims to provide a comprehensive understanding of the potential of magnetite-included 1D carbon nanostructure hybrids ffor EMI immunity by delving into the relevant perspectives and challenges. 

The novelty of research and application of hybrid composites based on one-dimensional carbon structures and nanomagnetite can be attributed to the combination and synergistic effect of the electrically conductive and magnetic properties of the components, which, when combined, result in an enhanced effect for protection against electromagnetic radiation. Furthermore, the combination of these properties allows for the formation of materials with universal shapes and the ability to function in a wide temperature range. The combination of these advantages, along with the variability of synthesis methods and their modifications, is of interest to researchers and reveals broad prospects for the creation of new hybrid composites.

This review analyses the strengths and possible challenges of various methods for synthesizing these materials and forming their hybrid structures. Special attention is paid to innovative applications of such hybrids, including their use in portable and high-performance electronic devices.

In recent years, there has been a notable increase in research interest in the development of multifunctional shielding materials that are capable of effectively shielding EMI without compromising functionality and minimizing environmental impact. However, to date, not enough research has been devoted to a comprehensive analysis of 1D carbon–magnetite hybrid nanostructures, their technological aspects of production, and specific applications in the field of EMI shielding. This review aims to fill this specific scientific gap by offering a new perspective on the application of innovative, environmentally friendly, and functionally efficient materials.

## 2. Fundamental Principles of Electromagnetic Shielding

The efficiency of electromagnetic interference shielding depends on the nature of the field source, and three basic types of shielding are distinguished: against electric, magnetic, and electromagnetic fields [26]. It is important to emphasize that electrostatic and magnetostatic shielding are specific cases of general electromagnetic shielding, although they are often considered separately because of the peculiarities of their interaction with the field [27].

The purpose of electrostatic shielding is to neutralize the effects of external static electric fields by redistributing charges on the surface of the conductor. This leads to a reduction of the field strength inside the shielded volume to zero, as in the case of using a grounded metal enclosure [28]. Static magnetic shielding, which aims to protect against the penetration of static magnetic and low-frequency magnetic fields, is usually achieved by using materials with high magnetic permeability, such as various ferromagnetic alloys [29]. Electromagnetic shielding combines methods of protection against electric and magnetic fields, using metals and magnetic materials to suppress the effects of both types of fields, which is particularly important for controlling the propagation of electromagnetic waves in sensitive areas.

Electromagnetic waves are defined by the presence of two components perpendicular to each other: a magnetic field (H) and an electric field (E), with waves propagating at right angles to the plane of these fields [30]. The wave impedance or impedance of such a wave (Z = |E|/|H|) varies with the source and distance from the source, stabilizing into a plane waveform at a certain distance from the source [31]. The near-field shielding efficiency is evaluated separately for magnetic and electric fields depending on their impedance [32]. The quantitative characterization of shielding effectiveness (SE) is based on the ratio of field strengths in the protected area of space in the absence of shielding (E_0_ or H_0_) and in the presence of shielding (E or H). The SE of a material can be expressed in simple ratios or decibels (dB).

It is important to acknowledge that there are instances where the efficacy of a screen is particularly difficult to ascertain. One such instance is when the area of the protected space in question is situated at a considerable distance from the screen itself. In addition, the worst point in the area and the worst possible location of the field source must be considered. In such situations, the accuracy of the estimate is significantly reduced, and researchers can only draw conclusions about the estimated minimum screening effectiveness based on the calculations performed.

Shielding mechanisms include reflection, absorption, and multiple reflection [33], as shown in Figure 2. Reflection is the primary mechanism by which interference is reflected from the surface of the shielding material [34]. Absorption depends on the properties of the material, such as its thickness and its ability to absorb electric or magnetic field components [35]. Multiple reflection occurs when interference is repeatedly reflected from the internal structures of a material, reducing the intensity of the transmitted interference [36].

As stated by the authors of the study [37], dielectric losses play a pivotal role in the process of electromagnetic wave absorption. The underlying mechanism of this process is absorption, which is dependent on the magnetic and dielectric losses within the material, as well as the direction of domains, walls, and resonance phenomena. In order to estimate these losses, it is necessary to determine the amount of electromagnetic waves incident on the absorber, which is reflected in the impedance matching coefficient. The incident electromagnetic wave penetrates the absorber, where it may either be scattered or converted to heat. By determining the impedance matching coefficient, the efficiency of electromagnetic wave absorption can be enhanced. 

Additionally, the principles of electromagnetic absorption can be influenced by the structure of the material, particularly porous structures, due to their high specific surface area and porous fiber composition. Electromagnetic waves undergo repeated scattering and reflection, ultimately converting into heat energy. This process contributes to more efficient absorption of electromagnetic waves. For example, the principles of electromagnetic absorption have been investigated using carbonized chitosan fibers obtained by electrospinning [38] as an illustrative example. This porous structure promotes the absorption of eddy currents, leading to an improved impedance matching coefficient.

Electromagnetic interference can occur over a wide range of frequencies, including radio frequencies and microwave frequencies. It has been demonstrated that as the level of electromagnetic interference shielding increases, the amount of energy transmitted through the shield is reduced. A shielding performance range of 10–30 dB is considered to be the minimum required for many applications [39]. Materials capable of providing an EMI protection level of 20 dB (equivalent to 99% EMI attenuation) are considered commercially acceptable and are widely used [40].

It should be noted that when discussing the general principles of electromagnetic shielding, the operating frequency range plays an important role. The same material may behave differently depending on the EMI frequency range [41]. Most papers deal with the effectiveness of EMI shielding in the X, Ku, and K bands [42,43,44]. The X band (8–12 GHz) is actively used in radar, air traffic control, satellite communications, and meteorology; the Ku band (12–18 GHz) is mainly used in satellite communications and extremely small aperture systems; and the K band (18–27 GHz) finds its application in radar systems [45].

## 3. Properties of 1D Carbon Nanostructures

This section details the features of carbon nanostructures such as carbon nanotubes (CNT) and carbon fibers (CFs), which are among the most promising materials for creating efficient composites for EMI shielding. These nanostructures have unique mechanical [46], electrical [47], and thermal properties [48] that make them ideal candidates for use in various applications, including EMI shielding.

### 3.1. Carbon Nanotubescar

CNTs are unique one-dimensional materials consisting of coiled graphene nanosheets, which are divided into single-walled carbon nanotubes (SWCNTs) and multi-walled carbon nanotubes (MWCNTs) [49]. Single-walled carbon nanotubes consist of a single layer of graphene and can have different structural shapes, such as zigzag, crescent, and chiral [50], while multi-walled carbon nanotubes are formed by folding several layers of graphene [51]. These structures exhibit outstanding physicochemical properties such as high mechanical strength, and excellent electrical and thermal behavior, making them attractive for a wide range of applications, including electromagnetic interference shielding, electronics, energy storage materials, and medical technology (Figure 3).

CNTs have been synthesized by various methods, including chemical vapor deposition (CVD), laser ablation, and arc discharge [52]. Each method has its own advantages and disadvantages. For example, CVD produces high-quality CNT with controlled dimensions, whereas arc discharge and laser ablation provide higher yields with potential structural defects. The quality and functionality of CNT are directly dependent on the chosen synthesis method, which highlights the importance of designing CNT manufacturing processes to achieve the specific properties required for different technological solutions [53].

The mechanical properties of CNT surpass the strength of steel by almost a hundred times with much lower weight [54], and due to sp^2^-hybridized carbon bonds, CNT can withstand temperatures of about 700 °C in air and up to 2800 °C in vacuum, indicating their outstanding electrical and thermal performance [55]. One of the main directions of CNT use is to improve the properties of polymer composites. The introduction of even a small amount of CNT into the polymer matrix significantly increases the mechanical strength and resistance of materials to mechanical stress [56]. Additionally, CNTs contribute to the improvement of the thermal and electrical conductivity of composites, which makes them attractive for use in electronics and power systems [57]. CNTs are also showing potential in the purification of water from various pollutants, the development of drug nanocarriers, contrast agents for medical imaging and biosensors, illustrating their wide range of applications and importance to various branches of science and technology [58,59]. The structural differences of SWCNT [57] and MWCNT [58], which affects their properties and potential in different fields, are summarized in Figure 4.

However, the widespread use of CNT is limited by several challenges, among which the uniform distribution of CNT in matrix materials, the environmental impact of CNT, and the scalability of production remain key issues [61]. CNT surface functionalization techniques aimed at improving their dispersion in solvents and compatibility with polymers can lead to defects or changes in the intrinsic properties of CNT, requiring a balance between functionalization and property preservation [62].

In the field of electromagnetic interference shielding, CNTs show promising results due to their high electrical conductivity and aspect ratio, which can effectively absorb and deflect electromagnetic interference [63]. However, synergistic interactions with various fillers increase the electromagnetic shielding efficiency of CNT, which shows improved performance compared to conventional materials [64]. Further research into synergistic combinations of CNT with other materials or nanostructures may lead to advances in achieving superior performance, including protection against electromagnetic interference.

### 3.2. Carbon Fiber

Carbon fibers are unique one-dimensional materials known for their exceptional strength, light weight, and high stiffness [65]. These synthetic fibers have found wide application in various industries, including aerospace, automotive, and sports manufacturing [66]. Their unique properties result from the conversion of carbon-containing polymers into highly ordered carbon structures through carbonization and graphitization processes [67]. The resulting materials, when woven into fabric or used to reinforce composites, provide excellent strength and stiffness while maintaining low weight. In addition, carbon fibers exhibit excellent resistance to corrosion and high temperatures, making them ideal for use in harsh environments. This combination of properties brings carbon fibers to the forefront of technological advances and engineering solutions [68]. Furthermore, in comparison to other carbon nanomaterials, such as carbon nanotubes and graphene, carbon fibers exhibit notable advantages [69]. For example, graphene is a single-layer carbon material arranged in a two-dimensional honeycomb lattice, whereas carbon fibers are one-dimensional structures with a high length-to-diameter ratio. This gives them unique mechanical properties and high specific surface area, which makes them particularly effective in composite materials [70].

The synthesis of carbon fibers begins with the selection of a suitable precursor. To date, the most widely used precursors are polyacrylonitrile (PAN), lignin, and carbon pitch [71]. Precursors undergo several processing steps, including fiber spinning, stabilization (oxidation), carbonization, and surface functionalization [72]. Fiber spinning is most commonly carried out by dry/wet spinning, melt spinning, or electrospinning [73], depending on the nature of the precursor (Figure 5). For example, PAN is usually processed by dry/wet spinning or electrospinning, whereas lignin or pitch can be processed by melt spinning, which can greatly simplify the process and reduce costs.

The main processing steps for precursor fibers include stabilization, carbonization, and activation (functionalization). Stabilization is a critical step where fibers are oxidized at 200–300 °C to convert them into heat-resistant materials. This process includes cyclization, dehydrogenation, and oxidation to ensure that the molecular and fibrillar orientation of the fibers is maintained. Stabilization is followed by carbonization, which is carried out at temperatures of 1000–2000 °C in an inert gas (N_2_) atmosphere. Carbonization converts the organic material into carbon fibers with a high carbon content of up to 95%. Surface functionalization of carbon fibers plays a key role in improving their properties and efficiency in end-use applications. Surface modification improves the interaction of the fibers with the composite matrix, resulting in materials with high mechanical and performance properties [74].

However, despite its high performance, carbon fiber production is associated with a number of challenges, especially in terms of economic and environmental sustainability [75]. The high energy consumption and greenhouse gas emissions associated with carbon fiber production cause environmental problems. Precursors such as PAN are associated with health risks to workers and the surrounding community [76]. It is of paramount importance to address the challenges associated with recycling carbon fiber-based materials [77]. Research is being conducted with the objective of enhancing recycling procedures and developing biodegradable composite materials incorporating carbon fibers in order to mitigate long-term environmental impacts [78].

Nevertheless, carbon fibers have a wide range of applications in various fields. In electronics and power systems, they are used to create high-performance electrodes in batteries and supercapacitors, which helps to increase the energy density and charging rate of devices [79,80,81]. In the automotive industry, carbon fibers are used to create lightweight and strong structures that help reduce energy consumption [82,83]. In aerospace and defense, carbon fibers are used to produce components with high mechanical performance and low mass [84,85]. Carbon fibers are also used in construction and other industries where their unique properties are important [86].

One of the most important applications of carbon fibers is protection against electromagnetic interference. According to their high electrical conductivity and excellent mechanical properties, carbon fibers are promising materials for use in electronic devices, aerospace systems, and telecommunications equipment. This is due to their ability to effectively shield electromagnetic waves.

## 4. Properties of Magnetite

Magnetite (Fe_3_O_4_) is a unique material with a cubic inverse spinel structure that provides high magnetic saturation and relatively low coercivity at room temperature. These properties make magnetite attractive for a wide range of technological solutions, including medicine, catalysis, and environmental technologies. Magnetite has a crystal lattice consisting of alternating layers of octahedrons and tetrahedrons, which gives it ferrimagnetic properties (Figure 6). Magnetite has a density of 5.18 g × cm^−3^, making it lighter than hematite but heavier than ferrihydrite. The electrical properties of magnetite vary between metallic and semiconducting with conductivity in the range of 102–103 ohm^−1^ × cm^−1^ and a forbidden bandwidth of about 0.1 eV [87,88]. 

The size of magnetite nanoparticles significantly affects their properties, including chemical activity and magnetic properties [92]. Nanoparticles have a high specific surface area, which increases their reactivity and allows surface functionalization for specific applications [93]. However, these same properties can lead to increased particle agglomeration without the use of stabilizers [94].

The main methods for the preparation of magnetite nanoparticles include coprecipitation, sol–gel, and solution combustion [89,90,91]. Coprecipitation is one of the most common and inexpensive methods, in which magnetite is precipitated from aqueous solutions of iron salts by changing the pH. This method is simple and inexpensive, but often results in particle size inhomogeneity and requires careful control of the reaction conditions to achieve the desired performance. The sol–gel method is based on the hydrolysis and condensation of precursors to form a sol, which is then gelled and dried to produce magnetite nanoparticles. This method produces particles with a high degree of homogeneity and controlled size but is time and resource consuming. Solution combustion is a method in which rapid combustion of organic substances in a metal ion-containing solution, resulting in the formation of magnetite nanoparticles. This method is characterized by high speed and low cost but can lead to agglomeration and requires optimized conditions to produce stable and homogeneous particles.

The magnetic properties of magnetite make it particularly valuable for applications requiring a fast and controllable response to external magnetic fields. In medicine, magnetite nanoparticles are used as contrast agents in magnetic resonance imaging (MRI), improving the quality of tissue and organ images [95]. Magnetite is also used in targeted drug delivery systems [96], where magnetite nanoparticles are modified to deliver drugs to specific areas of the body. In environmental engineering, magnetite is used for high-gradient magnetic separation (HGMS) to remove contaminants from water [97]. Magnetite nanoparticles can bind to suspended particles and precipitate as sludge, which can then be efficiently separated and recycled using a magnetic field [98]. The electrically conductive properties of magnetite are also of interest for use in composite materials for electromagnetic shielding applications [99]. However, it is important to note that a high degree of oxidation can degrade these properties, which requires precise control of the process conditions.

Despite their many potential applications, magnetite nanoparticles face challenges related to their toxicity and environmental impact. The adverse effects of magnetite nanoparticles on human health are significant; for example, they may cause neurodegenerative diseases due to their ability to aggregate with amyloid plaques in the brain [100,101]. Uncertainties about long-term sustainability and recyclability require further research and development of new technologies to minimize environmental impact.

## 5. Hybrid Composites Based on 1D Carbon Nanostructures and Magnetite

Hybrid composites incorporating 1D carbon nanostructures with magnetite have attracted considerable attention in recent years due to their promising potential to provide effective protection against electromagnetic interference. This unique combination of carbon nanostructures and magnetite gives the composites properties that are highly desirable for a wide range of applications, including aerospace, electronics, and telecommunications. This section reviews recent advances in the design, fabrication, and characterization of these hybrid composites and explores their potential impact on electromagnetic interference shielding. It also discusses the fundamental mechanisms that govern the electromagnetic behavior of these composites and highlights key challenges and future prospects in this dynamic area of research.

### 5.1. Magnetite/Carbon Nanotubes 

The unique properties of carbon nanotubes, such as high conductivity, light weight, mechanical strength, and ease of formation, make them interesting and promising materials for various applications. Despite this, carbon nanotubes in their native state have low efficiency in absorbing electromagnetic waves due to poor dispersion and insufficient magnetic properties. However, Fan Yang et al. [102] have shown in their work that a one-step method to fabricate dense carbon nanotube films leads to an increase in their electrical conductivity and electromagnetic interference shielding efficiency. Dense CNT films with a thickness of 14.3 μm have a conductivity of about 106 S × m^−1^ and a shielding efficiency of up to 71 dB at frequencies from 8.2 to 12.4 GHz. However, decorating CNT with Au particles helps to increase the conductivity up to 2.31 × 106 S × m^−1^ and improves the shielding up to 66.12 dB at a lower thickness of 3.3 μm. A significant number of researchers have dedicated their efforts to investigating the potential synergistic effects of CNT when combined with various dielectric and magnetic fillers in hybrid composites. This research aims to enhance the electrical conductivity and shielding properties of these materials. The development of hybrid materials comprising magnetite and carbon nanotubes represents a promising avenue for the creation of composite materials with the capacity to protect against electromagnetic interference. The unique properties of magnetite and CNT synergistically enhance the electromagnetic shielding efficiency of composites by providing magnetic properties, high electrical conductivity, and mechanical strength. 

Nowadays the authors have used various methods to obtain hybrid composites, among which solvothermal synthesis, in situ growth, polymer packing, electrostatic self-assembly (Figure 7), and classical deposition methods (Figure 8) stand out. It is worth noting that the choice of method has a direct impact on the structure of the resulting film, determining the size, distribution, interfacial interactions, dispersion, and agglomeration of the particles, as well as the thickness and shape of the coating.

**Figure 7 nanomaterials-14-01291-f007:**
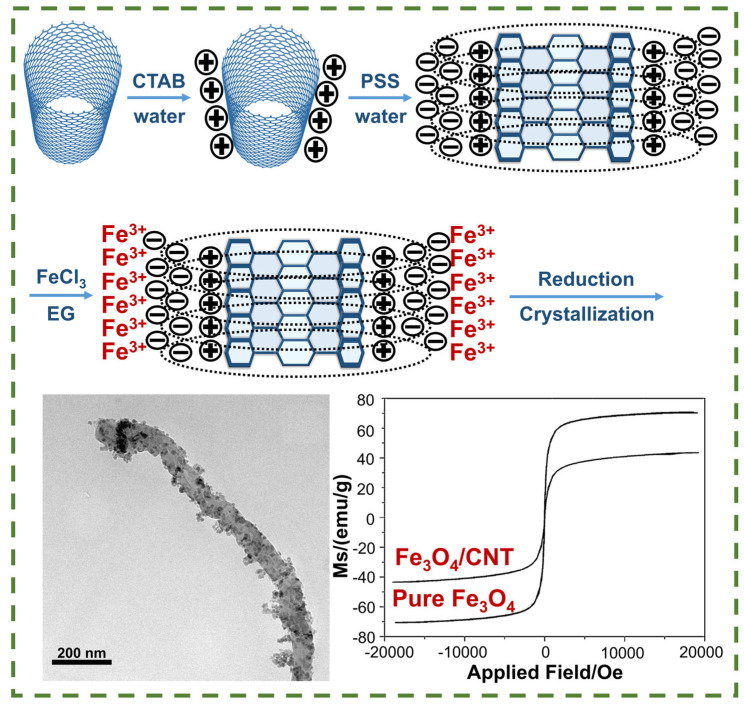
Preparation of Fe_3_O_4_/CNT composite and its magnetized saturation [103].

For example, Xuzhen Wang et al. [104] used an in situ solvothermal method to synthesize Fe_3_O_4_/CNT nanocomposites, which resulted in nanocomposites with a uniform distribution of quasi-spherical magnetite nanoparticles of size 10–25 nm on the outer surface of CNT due to the control of the initial mass ratio of ferrocene to CNT and the reaction temperature. It is worth noting that the choice of solvent and temperature profile critically affects the composite structure, such that reaction at 500 °C in benzene leads to the formation of small and uniform nanoparticles, whereas reaction in acetone or ethanol leads to the formation of Fe_3_O_4_ microparticles and carbon dendritic structures.

Similarly, in [105], Yingqing Zhan et al. state that in the synthesis of Fe_3_O_4_/CNT composites by solvothermal method, the particle size can be controlled by varying the concentration of the precursor (FeCl_3^−^_6H_2_O), and as a result, it is observed that by increasing the particle concentration from 1:1 to 8:1, the nanoparticle size increases from 10 to 100 nm. On the other hand, in [106], which describes a MWCNT/Fe_3_O_4_ nanocomposite prepared by hydrothermal synthesis, the authors highlight that the obtained nanocomposite was characterized by a dense and uniform coating of MWCNT with magnetite nanocrystals of about 8 nm in size. The composite had high saturation magnetization (28.67 emu/g) and significant microwave absorption in the 2–18 GHz range with a minimum reflection loss of −41.61 dB at 5.5 GHz. However, Yong Liu et al. [103] were able to achieve a higher saturation magnetization (43.5 emu × g^−1^) by using a polymer wrapping technique and electrostatic self-assembly of the composite, as shown in Figure 7. The high magnetization of the material in turn contributes to the absorption and reduced reflection of electromagnetic waves through magnetic loss and interface polarization mechanisms. 

**Figure 8 nanomaterials-14-01291-f008:**
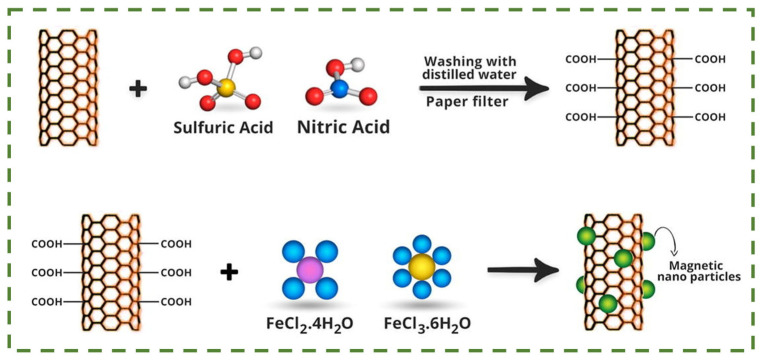
Preparation of Fe_3_O_4_/CNT composites by deposition method [107].

T. Zhang et al. [108] also describe the effect of soaking carbon nanotubes in a gel containing magnetite nanoparticles. It was demonstrated that the morphology of the composite is directly dependent on the soaking time of the CNT; that is, with increasing soaking time, the surface of the nanotubes becomes increasingly rough due to the increase in the number of magnetite nanoparticles. The composite exhibits an average size of magnetite nanoparticles (approximately 30 nm) and a high microwave absorption capacity in the range of 2–18 GHz, accompanied by minimal reflection loss up to −51.2 dB at 11.7 GHz. 

Conversely, Na Li et al. [109] describe the influence of the type of coating of carbon nanotubes with magnetite nanoparticles. Two distinct structures were obtained by the solvothermal method: compact (FCC) and friable (FLC). The FCC structure provides a dense distribution of small magnetite nanoparticles on the surface of carbon nanotubes, while the FLC structure is characterized by larger and less densely distributed nanoparticles. Consequently, the compact-coated composite exhibited a minimum reflection loss (RL) of −43 dB and a bandwidth of 8.3 GHz at a thickness of 1.5 mm. In contrast, the loose coating exhibited less effective absorption due to the lower coating density and larger size of the nanoparticles. The achievement of such values can be facilitated by the implementation of higher coating density and smaller nanoparticle size, which serves to enhance impedance matching and interfacial polarization. It is of great importance to control the distribution and density of magnetic nanoparticles on the CNT surface in order to optimize microwave absorption. A study by Kang Zhang et al. [110] also demonstrated a novel approach to improve microwave absorption performance by tuning the defect sites in composites through direct fluorination (Figure 9). Two types of fluorinated MWCNTs were prepared: surface-fluorinated MWCNTs (S-F-MWCNTs) and deeply fluorinated MWCNTs (D-F-MWCNTs), each with a different defect distribution. The S-F-MWCNT/Fe_3_O_4_ composites exhibited defects primarily on the outer tubes, whereas the D-F-MWCNT/Fe_3_O_4_ composites exhibited defects on both the inner and outer tubes. The strategic placement of defects significantly increased the polarization and conduction losses in the electromagnetic field, resulting in superior microwave absorption for D-F-MWCNT/Fe_3_O_4_ compared to S-F-MWCNT/Fe_3_O_4_.

In a study [111], a novel necklace-like nanocomposite based on Fe_3_O_4_ and carbon nanotubes was synthesized using a one-step hydrothermal method (Figure 10). The synthesis resulted in the formation of homogeneous spherical Fe_3_O_4_ nanoparticles that grew along the walls of CNT, thereby forming a three-dimensional necklace-like structure. It is noteworthy that such composites exhibit excellent magnetic and dielectric properties, with a minimum RL value of −59.2 dB at a thickness of 1.68 mm. Furthermore, these composites exhibit high electrical conductivity and rapid surface redox reactions, conferring excellent electrochemical properties (specific capacitance of 361.1 F × g^−1^ at 0.5 A × g^−1^). The authors demonstrated that a successful combination of physical and chemical properties can be realized in a single composite structure, which not only improves material utilization but also enables the creation of multifunctional devices.

Zeng et al. [112] employed a solvothermal process to synthesize Fe_3_O_4_ nanoflower-CNT composites, resulting in a minimum reflection loss of −58.6 dB at 15.28 GHz with a thickness of only 1.52 mm. The excellent microwave interference absorption by the composites can be explained by the effective dielectric loss control and significant interfacial polarization between Fe_3_O_4_ and CNTs. In another study, Mohammad Shamsaddin Saeed et al. [113] enhanced MWCNTs with Fe_2_O_3_/Fe_3_O_4_ nanoparticles to enhance microwave absorption. Furthermore, Zhijiang Wang et al. [114] employed a chemoselective synthesis approach to create MWCNT/Fe_3_O_4_@ZnO heterotrimers, which resulted in the formation of multiple interfaces within the hybrid structure. The strategic formation of C-O-Zn, C-O-Fe, and Fe-O-Zn bonds resulted in enhanced microwave absorption throughout the X band (8–12 GHz), with heterotrimers outperforming binary hybrids such as MWCNT/Fe_3_O_4_ or MWCNT/ZnO due to their superior electron charge redistribution and interfacial engineering. Both studies emphasize that the combination of high magnetic permeability and dielectric loss leads to improved impedance matching and superior microwave attenuation capabilities. On the other hand, Tirtharaman Govindasamy et al. [115] manufactured polymer-free standalone Fe_3_O_4_/MWCNT nanocomposites, achieving an average shielding efficiency of 60.7 dB in the X-band region with a thickness of only 500 μm. This high performance is attributed to the presence of Fe-O-C bonds, which enhance interface polarization, electron charge transfer ability, and impedance matching. It is important to note that the thickness of the composite plays a significant role in determining its electromagnetic performance. Thinner composites may be favored due to their lightness, flexibility, and cost-effectiveness. 

For example, Wenlai Xia et al. [116] in their study obtained composites with a minimum reflection loss of −52.57 dB at 3.3 mm thickness and a maximum effective absorption bandwidth of 5.2 GHz at 1.4 mm thickness. Additionally, Wen-wu Jin et al. [117] recently achieved the synthesis of three-component CNT/MCF/Fe_3_O_4_ composites through the novel integration of metal–organic frameworks (MOFs) with carbon-based skeletons (Figure 11). The obtained composites exhibited an overall shielding efficiency of 46.41 dB in the X band at a thickness of 3 mm and demonstrated a satisfactory degree of composite stability following compression cycles (SET 33.80 dB after 50 cycles).

The collective findings of these studies underscore the pivotal role of synthesis techniques and structural configurations in optimizing the microwave absorption properties of magnetite-CNT composites. The outcomes demonstrate that strategic assembly and interfacial engineering, encompassing the formation of multiple interfacial bonds and the control of nanoparticle distribution, are indispensable for the advancement of high-performance microwave absorbers. Future research should continue to explore new synthesis methods and hybrid configurations in order to further improve the electromagnetic properties of these composites for practical applications in electromagnetic interference protection and other advanced technologies. Table 1 provides a summary of the results on the key electromagnetic properties of magnetite and carbon nanotube hybrid structures obtained by different methods.

### 5.2. Magnetite/Carbon Fibers

Carbon fibers are employed extensively in electromagnetic interference protection due to their pronounced dielectric sensitivity, high tensile strength, heat resistance, and light weight [123]. In contrast to carbon nanotubes, carbon fibers exhibit a variable aspect ratio, which allows the conductivity of the material to be adjusted. The conductivity of the material is critical for effective impedance matching, which makes carbon fibers an equally promising material in the field of electromagnetic shielding. Nevertheless, primary carbon fibers do not have high values in the attenuation of electromagnetic interference, which is directly related to their low magnetization [124]. In this regard, carbon fibers undergo various structural and morphological modifications through thermal or chemical actions [125]. 

The authors of [126] investigated the potential application of carbon fiber mats derived from PAN with different additions of zirconium oxide (ZrO_2_) as a modifying additive in order to be used for electromagnetic interference protection (Figure 12). Consequently, carbon fibers and composite fibers with varying diameters, spanning from 4 to 8 μm, were obtained. The authors also investigated the effect of zirconium dioxide introduction on the electrical conductivity of the fibers. It is worth noting that electrical conductivity is one of the key factors affecting protection against electromagnetic interference. It is known that the conductivity of the material contributes to the effective reflection and absorption of waves. Reflection is due to the ability of free electrons in the conducting material to resist the electromagnetic field, returning energy back to the surrounding space. Absorption, on the other hand, is due to the interaction of the electromagnetic field with the electrons, converting some of its energy into heat. 

Furthermore, the microstructural properties of nanofibers can result in the scattering of waves, which subsequently reduces the energy of the waves. The authors [126] discovered that pure carbon fibers exhibit AC conductivity in the range of 0.0035 S × cm^−1^, and when they are modified with zirconium oxide particles, the conductivity increases by a factor of three. However, Ji Sun Im et al. [127] report that pure PAN-based carbon fibers have a conductivity of about 4.5 S × cm^−1^, which is much higher than the results of the previous study. The difference in the results may be the type of current applied in the conductivity study; thus, B.D.S. Deeraj et al. [126] used alternating current, whereas Ji Sun Im et al. used direct current. On this basis, it can be assumed that the type of current applied can significantly influence the results of conductivity measurements of a material. The difference between direct current (DC) and alternating current (AC) in the context of conductivity measurement is a key aspect to consider. To illustrate, when measuring conductivity using direct current, the current flows through the sample in a single direction, allowing the resistance or conductivity of the material to be quantified without the influence of current frequency. Conversely, the utilization of alternating current in conductivity measurements introduces an additional layer of complexity due to the influence of current frequency. This phenomenon can result in a variation in the conductivity of a material at different frequencies as a consequence of the skin effect, whereby the current is forced to the surface of the conductor at high frequencies. Despite this, alternating current allows the dielectric properties of the material, such as dielectric constant and dielectric relaxation loss, to be studied, which is important for understanding the behavior of the material in alternating electromagnetic fields. Accordingly, the authors found that carbon fibers had an average shielding efficiency of −7.7 dB, while modified ZrO_2_ had shielding efficiencies ranging from −9.8 dB to −15.6 dB. The absorption of electromagnetic waves represents the primary shielding mechanism. The combined action of carbon fibers and zirconia results in a high dielectric constant and dielectric loss, which is also confirmed by AC conductivity measurements. 

On the other hand, a recent study [128] combined nanomembranes containing cobalt and iron, which exhibited high microwave absorption efficiency and significant antibacterial and water—repellent properties (Figure 13). Among the samples studied, the CNFs 0.6-Co@Fe_3_ composite exhibited the most favorable performance, with a minimum reflection of −29.10 dB at a thickness of 1.5 mm in the frequency range of 15–20 GHz. This phenomenon can be attributed to the rise in the concentration of nanoparticles within the matrix, which results in an enhancement of the complex dielectric permittivity and magnetic permeability of the composites. It is crucial to highlight that the augmentation in PVA content exerts a constructive influence on the RL performance, which is attributable to the elevation in solution viscosity and enhanced conductivity.

A study [129] described a method for the fabrication of Fe/Fe_3_O_4_/C-based fibrous nanocomposites obtained using kapok fibers as a template (Figure 14). In the fabrication process, kapok fibers were impregnated with Fe(NO_3_)_3_ solution and subjected to argon firing, resulting in the formation of Fe/C or Fe_3_O_4_/C fibers. Considerably, this simple modification of the precursor fibers allows for obtaining a rough surface on which Fe or Fe_3_O_4_ grains can be easily anchored. The study of electromagnetic parameters revealed that fibers with a low precursor concentration exhibited higher values of the real and imaginary parts of the complex permittivity (ε′ and ε″) in the high frequency range (10–18 GHz), indicating significant dielectric relaxation and losses. The magnetic properties were evaluated using a vibrating magnetometer (VSM), which revealed high values of saturation magnetization and low coercivity, which are characteristic of soft magnetic materials, while fibers with high precursor concentration exhibited lower saturation magnetization values. This may be attributed to the higher content of Fe_3_O_4_ nanoparticles, which exhibit a lower magnetization compared to Fe. In addition, the low precursor concentration fibers provided a wider effective absorption range with a minimum RL value of −40.1 dB at a thickness of 5.5 mm and a frequency of 2.2 GHz. It can also be suggested that, despite the relatively large thickness of the composites, the hollow structure of the material may adversely affect the mechanical strength of the composites.

Conversely, Weihong Zhou et al. [130] describe a method for the synthesis of CoFe_2_O_4_@PCF nanocomposites based on porous carbon fibers derived from silkworm cocoons through a degumming and firing process. It is of interest to note that in both cases [129,130], the use of natural precursor fibers contributed to the production of composites with high surface area and improved electromagnetic properties. It is also of interest to note that composites with a low concentration of CoFe_2_O_4_ have higher values of the real and imaginary parts of dielectric permittivity (ε′ and ε″), which is similar to the results of [129]. However, the magnetic properties of the composites investigated by Weihong Zhou et al. demonstrated a higher saturation magnetization (M_s_) for those with a high CoFe_2_O_4_ content. Furthermore, the electromagnetic wave absorption efficiency exhibited a minimum RL value of −53.3 dB at a thickness of 2.63 mm at 7.17 GHz, which represents a superior outcome to that observed in previous studies (Figure 15).

In a further recent study [131], complex hybrid multicomponent nanocomposites have been developed which exhibit high mechanical and electromagnetic performance due to synergistic interactions. The 0.2% GO-0.2% MP-CFPPy-Epoxy composite exhibited the most favorable results in terms of electromagnetic shielding and mechanical performance, with a shielding efficiency of 46.72 dB in the frequency range of 8–12 GHz. The authors note that due to the interaction between graphene oxide and magnetite, the tensile strength and Young’s modulus increase. On the other hand, in [132], higher mechanical and electromagnetic properties were achieved due to the multilayer structure of the CF_f_/GO/Fe_3_O_4_/epoxy composite. The maximum tensile strength value was 565.1 MPa, which is much higher than other composites described earlier. The EMI shielding efficiency reached 32.9 dB, which was due to both dielectric and magnetic losses. 

Sateesh Bandaru et al. [133] obtained a nanocomposite based on ferrite and carbonized cotton fibers (CCFs-Fe_3_O_4_) obtained by carbonization of cotton fibers at different temperatures and processing times. The study describes the effects of temperature and time periods on the structural and electromagnetic properties of the composite. Carbonization at 800 °C for 80 min yields composites with the highest values of the real and imaginary parts of the complex permittivity (ε′ and ε″) in the frequency range of 2–18 GHz, while the highest values of saturation magnetization of the composite were achieved at a maximum exposure time of 120 min due to the higher content of Fe_3_O_4_ nanoparticles. Nevertheless, the samples carbonized for 80 min showed higher magnetic loss capability and better electromagnetic wave absorption efficiency. The minimum RL value was −56.8 dB at 10.9 GHz and 1.67 mm thickness, indicating high absorption performance. It is also noteworthy that the sample exhibiting the highest saturation magnetization result exhibited a lower result in electromagnetic protection due to the agglomeration of Fe_3_O_4_ nanoparticles, which reduces the efficiency of their distribution and interaction with the matrix, reducing the interaction surface between nanoparticles and electromagnetic waves. The high concentration of nanoparticles leads to non-uniform distribution, which deteriorates impedance matching and increases the reflection of electromagnetic waves on the material surface, reducing the overall absorption efficiency. 

In a study by Govind Kumar Sharma et al. [134], the effect of composite thickness on electrical conductivity and (EMI) shielding efficiency was investigated. The results demonstrated that composite (PEDOT:PSS-PVP/CNF) at thicknesses of 0.04 and 0.06 mm has shielding effectiveness (SE) values of 25.9 and 43.4 dB, respectively. It can be concluded that the thickness of the composite influences its electrical conductivity, with an improvement in conductivity resulting from the creation of more conductive pathways. However, it is important to note that the final effect is dependent on the properties of the conductive fillers added and their distribution within the polymer matrix. For instance, Veerendra Kumar Patle et al. [135] prepared Ni- and Fe-decorated carbon fiber composites using an identical method. However, the Fe-decorated composite exhibited EMI shielding efficiency that was 13% higher than that of the Ni-based composite. This may be attributed to the superior magnetic properties of iron, which facilitate the efficient absorption and scattering of electromagnetic waves, transforming their energy into heat. Furthermore, the higher electrical conductivity of iron compared to nickel enhances the shielding ability, creating a more reliable protective barrier. Presumably, the distribution of metallic nanoparticles in the carbon fiber structure may also play a role. These factors collectively explain the superiority of iron composites over other metal fillers in electromagnetic shielding applications.

The analysis of the studies reveals the pivotal role of synthesis methods and structural configurations in optimizing the electromagnetic wave absorption properties of Fe_3_O_4_-based nanocomposites and carbon fibers. The findings indicate that strategic nanoparticle distribution and enhanced interfacial interaction, such as the formation of porous structures and the control of nanoparticle size and shape, are crucial for the development of highly efficient electromagnetic wave absorbers. Future research should continue to explore new synthesis approaches and hybrid configurations to further improve the electromagnetic performance of these composites. This will allow for more applications in electromagnetic interference protection and other high-tech fields. Table 2 provides a summary of the key electromagnetic properties of Fe_3_O_4_-based nanocomposites and carbon fibers prepared by different methods. 

### 5.3. Magnetite/Carbon Nanorods

Carbon nanorods (CNs) are infrequently incorporated into composites with magnetite for the purpose of electromagnetic interference (EMI) shielding. Noteworthy methods explored for synthesizing such composites during the period 2008–2018 include ultrasonic treatment [145], pyrolysis [146], hydrothermal synthesis of magnetite/C nanorods [147,148], and calcination [149], primarily aimed at their application in supercapacitors and lithium-ion batteries. However, CNs can also be utilized for developing hybrid materials aimed at EMI shielding, leveraging their distinctive geometric structure and high dielectric loss attributable to their diminutive size and substantial specific surface area [150,151].

Wang et al. [152] synthesized one-dimensional composites of core/shell Fe_3_O_4_/carbon nanotubes through a process involving magnetic-field-induced synthesis followed by high-temperature carbonization. The study compared Fe_3_O_4_/C core/shell nanoparticles with Fe_3_O_4_/C core/shell nanorods and found that the latter exhibited superior dielectric properties and higher reflection loss in a high frequency range when used as thin absorbers. Furthermore, the study demonstrated that the core/shell nanorod structure enhances both electrical energy storage capacity and loss capability, exhibiting higher dielectric losses compared to Fe_3_O_4_/C core/shell nanoparticles. This enhancement is attributed to the synergistic effects of dielectric and magnetic losses arising from increased interfacial polarization and a larger aspect ratio of the core/shell structure in Fe_3_O_4_/C nanorods. Consequently, these nanorods displayed excellent microwave absorption characteristics, achieving an optimal peak absorption value of −44.1 dB at 12.8 GHz with a thickness of 1.6 mm. The bandwidth, where the reflection loss remained below −10 dB, extended over 2.7 GHz from 11.6 GHz to 14.3 GHz. These findings underscore the potential of Fe_3_O_4_/C core/shell magnetic nanorods as promising candidates for electromagnetic interference (EMI) shielding applications.

Jia et al. [153] synthesized an expanded graphite/Fe_3_O_4_/CN composite using microwave puffing at 1000W. The carbon nanorods served to bind the expanded graphite/Fe_3_O_4_ particles into a cohesive structure, enhancing the synergistic effects with graphene. This resulted in the formation of a three-dimensional conductive network facilitated by the CN, thereby improving the electrical conductivity and dielectric properties of the nanocomposite. The researchers observed that ferrocene, employed as a growth catalyst, underwent decomposition to form magnetic particles, thereby enhancing the absorption properties of the composite. The resultant material exhibited favorable relative dielectric constant and magnetic conductivity characteristics. The analyses indicated that at a thickness of 2.4 mm, the carbon nanocomposite achieved a minimum absorption peak of −28 dB at 14.4 GHz, with a frequency bandwidth below −10 dB that could reach 5.5 GHz from 2 GHz to 18 GHz. Variations in the composite’s dielectric permittivity with frequency variations further underscored the magnetic absorption capabilities of the expanded graphite/Fe_3_O_4_/CN composite material.

These investigations highlight the potential of CNs in applications related to EMI shielding and provide a basis for advancing CN-based materials for this purpose. The distinctive aspect of magnetite/CN composites lies in the ability to manipulate the mass ratio and morphology of magnetic particle distribution, thereby controlling the electromagnetic parameters of the nanocomposites. This control enables the optimization of impedance matching characteristics by adjusting the ratio between complex dielectric constant and complex permittivity. Moreover, the structural features of CNs significantly influence their dielectric properties, underscoring their critical role in material performance for EMI shielding applications [153].

### 5.4. Magnetite/Carbon Nanowires

In 2017, Chen et al. [154] conducted a study where Fe_3_O_4_ nanocrystals were combined with nitrogen-doped carbon nanowires through calcination, aimed at enhancing the performance of supercapacitors. However, subsequent research addressing the synthesis and practical use of Fe_3_O_4_/Carbon nanowire hybrids, especially in the context of EMI shielding, has not been documented. Although carbon nanorods and nanowires exhibit good electrical conductivity, their conductivity may be inferior to metallic materials like copper or aluminum. This limitation can lead to reduced shielding effectiveness, particularly at higher frequencies. Additionally, the compatibility of magnetite nanorods with carbon in composite materials is noted to be superior compared to the compatibility of magnetite with carbon nanorods, primarily due to the relatively weaker synergistic effects observed in the latter case. These factors are considered to be the primary reasons for the limited adoption of hybrids combining magnetite with carbon nanorods and nanowires in practical applications. Nevertheless, high surface area, light weight, mechanical strength, flexibility, malleability, and chemical resilience of carbon nanorods and nanowires [155,156] indicate promising potential for their integration with magnetite in composite materials, particularly for EMI shielding. This potential warrants further comprehensive investigation and analysis, representing an open area for future research.

## 6. Conclusions and Prospects

The development of efficient electromagnetic interference shielding materials, characterized by high absorption capacity, broadband absorption, and low density, is crucial to reduce electromagnetic pollution and enhance microwave absorption capabilities. This review summarizes advances in the development of hybrid composites based on one-dimensional carbon nanostructures and magnetite, demonstrating their potential as efficient electromagnetic absorption materials.

Carbon nanotubes and carbon fibers combined with Fe_3_O_4_ show significant promise in improving electromagnetic interference shielding efficiency. However, achieving uniform distribution of these nanostructures in polymer matrices remains a challenge due to van der Waals forces causing agglomeration and impedance mismatch. To address this issue, surface functionalization, and binary fillers can be used to improve dispersion and interaction between the matrix and fillers. Moreover, surface modification with ceramics and semiconductors can improve the thermal stability and oxidation resistance of CNT and CFs, making them more suitable for high-temperature applications.

Despite these advances, the practical application of these materials faces a number of challenges. The production methods for CNT and CFs are often complex and expensive, so it is important to scale these processes while maintaining homogeneous quality and reducing production costs. In addition, the mechanical stability of multilayer structures can be compromised by cracking, limiting their practical use. Future research should focus on optimizing synthesis processes and exploring new hybrid configurations to further improve the electromagnetic properties of these composites. The use of renewable carbon and activated carbon can provide additional benefits such as weight reduction, corrosion resistance, and environmental friendliness.

With the rapid development of technology, there is an increasing demand for multifunctional EMI protection materials that can operate in multiple frequency ranges, including infrared, laser, X-ray, and ultraviolet waves. In this context, hybrid structures that provide EMI protection over a wide range of frequencies represent a promising direction. Understanding the fundamental factors that determine the effective absorption bandwidth and optimum thickness of these materials is crucial for their optimization. In addition, studying the microscopic phenomena associated with microwave absorption of Fe_3_O_4_, such as molecular orbital quantum numbers and energy level transitions, can provide valuable information for the improvement of these materials.

The versatility of hybrid composites based on one-dimensional carbon nanostructures and Fe_3_O_4_ is a significant factor. In addition to their excellent EMI shielding capabilities, these materials can possess other beneficial properties, including enhanced mechanical strength, thermal stability, and environmental resistance. For instance, hybrid composites can exhibit self-healing properties, improved thermal conductivity, and even photocatalytic activity. The additional functionalities of hybrid composites render them highly versatile and suitable for a wide range of advanced applications, including aerospace, electronics, and environmental protection.

In conclusion, Fe_3_O_4_-based hybrid composites with one-dimensional carbon nanostructures have significant potential for next-generation electromagnetic interference protection. Permanent research and innovation in this field are required to fully utilize the potential of these materials and to ensure that they satisfy the practical and industrial requirements of new electromagnetic absorption technologies.

The development of efficient electromagnetic interference (EMI) shielding materials, characterized by high absorption capacity, broadband absorption, and low density, is crucial to reduce electromagnetic pollution and enhance microwave absorption capabilities. This review summarizes advances in the development of hybrid composites based on one-dimensional carbon nanostructures and magnetite, demonstrating their potential as efficient electromagnetic absorption materials. 

Carbon nanotubes (CNTs) and carbon fibers (CFs) combined with Fe_3_O_4_ show significant promise in improving electromagnetic interference shielding efficiency. However, achieving uniform distribution of these nanostructures in polymer matrices remains a challenge due to van der Waals forces causing agglomeration and impedance mismatch. To address this issue, surface functionalization and binary fillers can be used to improve dispersion and interaction between the matrix and fillers. Moreover, surface modification with ceramics and semiconductors can improve the thermal stability and oxidation resistance of CNTs and CFs, making them more suitable for high-temperature applications.

Despite these advances, the practical application of these materials faces a number of challenges. The production methods for CNTs and CFs are often complex and expensive, so it is important to scale these processes while maintaining homogeneous quality and reducing production costs. Additionally, the mechanical stability of multilayer structures can be compromised by cracking, limiting their practical use. Future research should focus on optimizing synthesis processes and exploring new hybrid configurations to further improve the electromagnetic properties of these composites. The use of renewable carbon and activated carbon can provide additional benefits such as weight reduction, corrosion resistance, and environmental friendliness. 

With the rapid development of technology, there is an increasing demand for multifunctional EMI protection materials that can operate in multiple frequency ranges, including infrared, laser, X-ray, and ultraviolet waves. In this context, hybrid structures that provide EMI protection over a wide range of frequencies represent a promising direction. Understanding the fundamental factors that determine the effective absorption bandwidth and optimum thickness of these materials is crucial for their optimization. In addition, studying the microscopic phenomena associated with microwave absorption of Fe_3_O_4_, such as molecular orbital quantum numbers and energy level transitions, can provide valuable information for the improvement of these materials. 

The versatility of hybrid composites based on one-dimensional carbon nanostructures and Fe_3_O_4_ is a significant factor. In addition to their excellent EMI shielding capabilities, these materials can possess other beneficial properties, including enhanced mechanical strength, thermal stability, and environmental resistance. For instance, hybrid composites can exhibit self-healing properties, improved thermal conductivity, and even photocatalytic activity. These additional functionalities render hybrid composites highly versatile and suitable for a wide range of advanced applications, including aerospace, electronics, and environmental protection. 

The materials reviewed in this manuscript show significant potential in several key applications. In electronics, flexible and lightweight composites such as PANI/MXene/CF fabric and WS_2_-CF composites are suitable for wearable electronics, providing protection against EMI without adding bulk or weight. Composites like Fe_3_O_4_/MWCNTs@CFs and MWCNT@Fe_2_O_3_/Fe_3_O_4_ can be used in smartphones, laptops, and other consumer electronics to protect sensitive components from electromagnetic interference. In the aerospace and automotive industries, high-performance EMI shielding materials such as CFf/GO/Fe_3_O_4_/Epoxy composites and Fe_3_O_4_/CF/cement composites can be used in the construction of aircraft and vehicles to shield electronic systems from electromagnetic interference and ensure their reliable operation. For medical devices, the EMI shielding properties of these composites can protect medical devices from external electromagnetic fields, ensuring their accuracy and reliability. Materials like the PANI@nano-Fe_3_O_4_@CFs/epoxy hybrid composite are particularly suited for this application. In telecommunications, advanced EMI shielding materials can be used in the construction of 5G infrastructure to minimize interference and enhance signal integrity. Composites like MWCNT/Fe_3_O_4_@ZnO are ideal candidates due to their excellent dielectric and magnetic properties.

However, there are several challenges that need to be addressed before these materials can be widely applied. Developing scalable manufacturing processes for these composites is crucial. Methods that ensure uniform dispersion of nanoparticles and maintain the integrity of the composite structure during large-scale production need to be optimized. Ensuring the long-term durability and stability of these composites under various environmental conditions is essential. Studies on the aging, weathering, and mechanical robustness of these materials will help in understanding their practical lifespan. Reducing the cost of raw materials and synthesis processes is necessary to make these composites economically viable for widespread use. Exploring alternative, less expensive materials and optimizing synthesis methods can help in achieving this goal. Ensuring that these materials comply with regulatory standards and safety guidelines is critical for their application in sensitive fields such as healthcare and aerospace. Developing methods to seamlessly integrate these advanced composites with existing technologies and systems is crucial for their practical implementation. Research on hybrid systems that combine traditional materials with advanced composites can facilitate this transition.

In conclusion, Fe_3_O_4_-based hybrid composites with one-dimensional carbon nanostructures have significant potential for next-generation electromagnetic interference protection. Ongoing research and innovation in this field are required to fully utilize the potential of these materials and to ensure that they satisfy the practical and industrial requirements of new electromagnetic absorption technologies.

## Figures and Tables

**Figure 1 nanomaterials-14-01291-f001:**
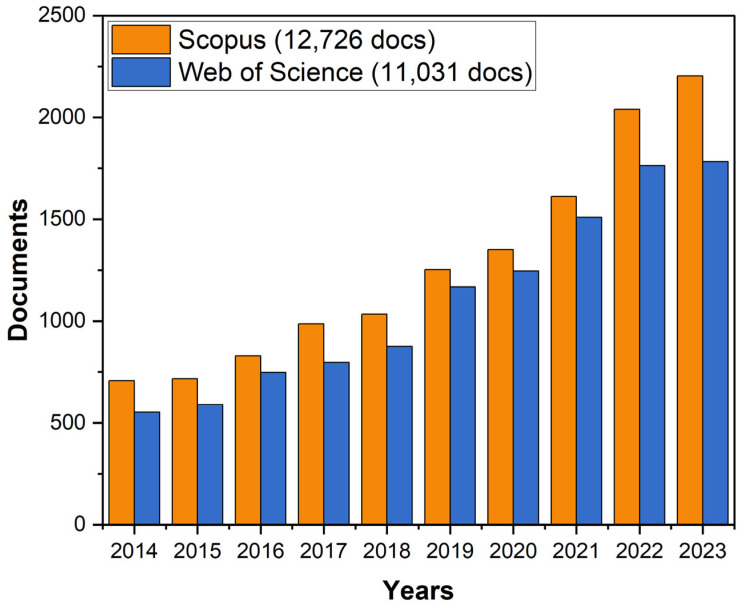
Scientific publications for the last 10 years in the field of EMI shielding materials according to Scopus and Web of Science databases.

**Figure 2 nanomaterials-14-01291-f002:**
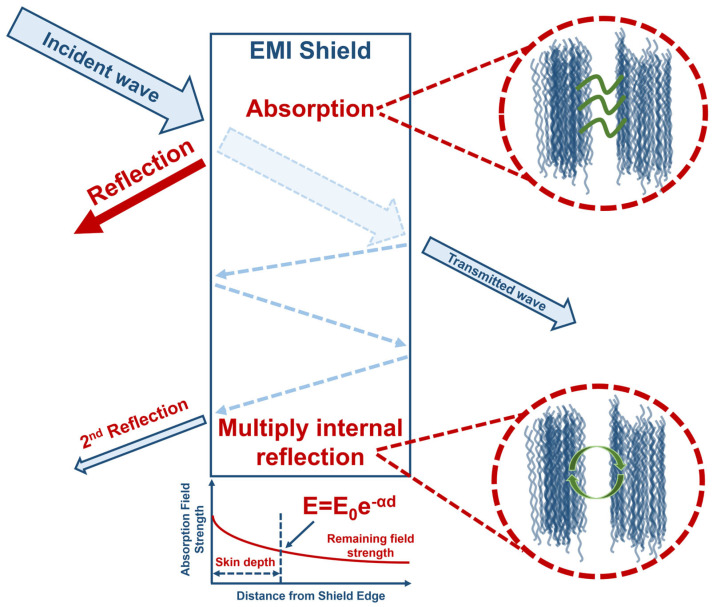
Schematic diagram of the electromagnetic interference shielding mechanism.

**Figure 3 nanomaterials-14-01291-f003:**
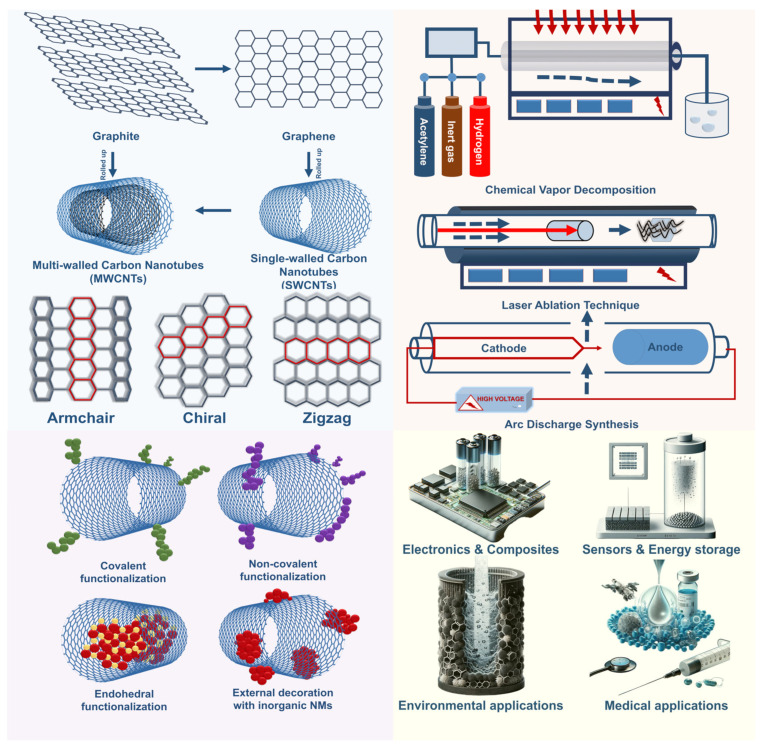
Structural features of carbon nanotubes, their methods of production, and actual fields of application.

**Figure 4 nanomaterials-14-01291-f004:**
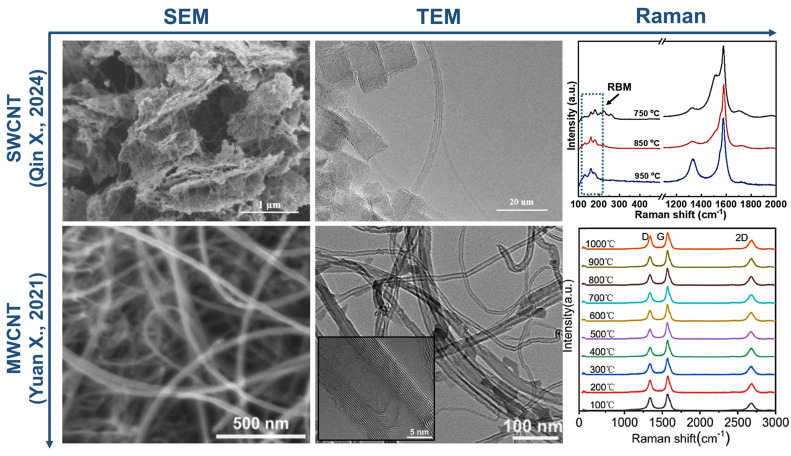
Characterization of carbon nanotubes by different methods [59,60].

**Figure 5 nanomaterials-14-01291-f005:**
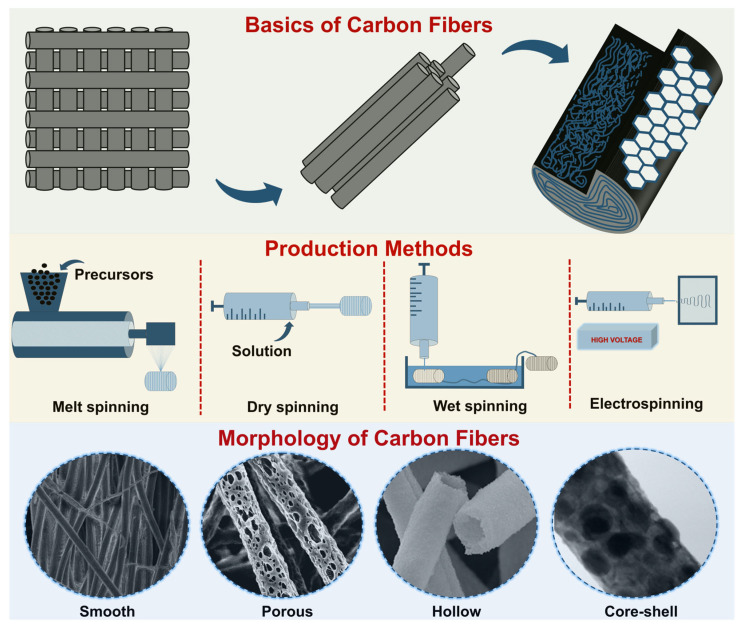
Structural features of carbon fibers and methods of their production.

**Figure 6 nanomaterials-14-01291-f006:**
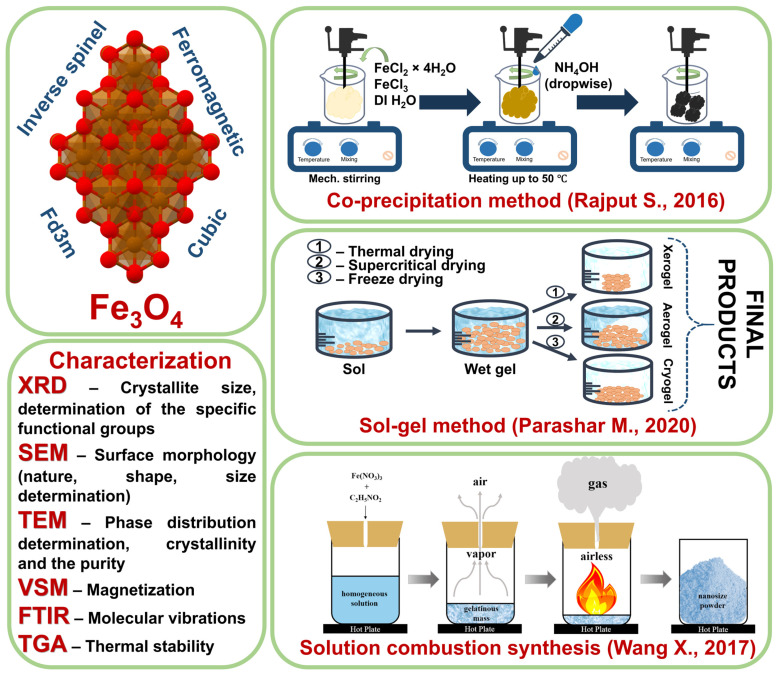
Methods of production [89,90,91] and characterization of magnetite.

**Figure 9 nanomaterials-14-01291-f009:**
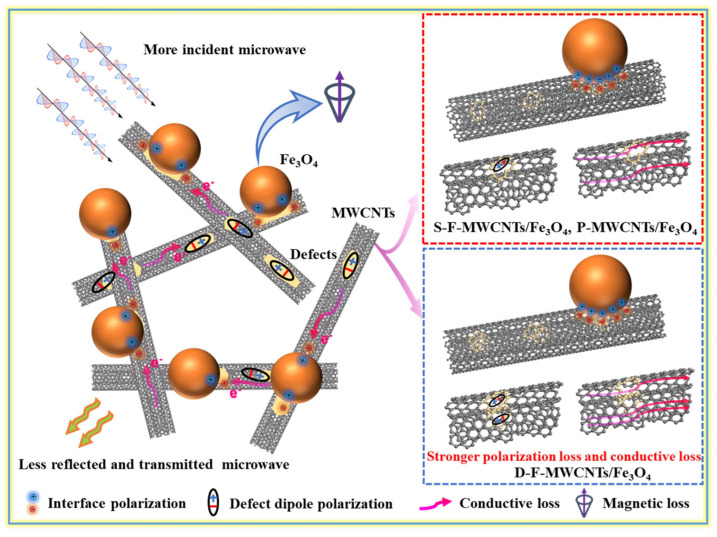
Schematic illustration of the microwave absorption mechanisms of the different composites based on CNT and Fe_3_O_4_ [110].

**Figure 10 nanomaterials-14-01291-f010:**
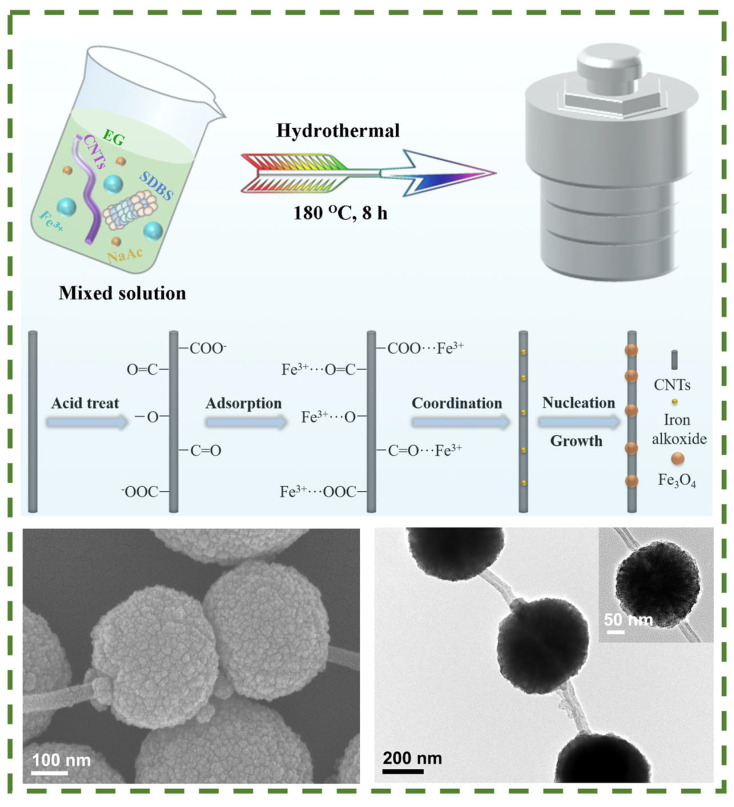
Preparation of necklace-like 3D Fe_3_O_4_/CNT composites by one-pot solvothermal method [111].

**Figure 11 nanomaterials-14-01291-f011:**
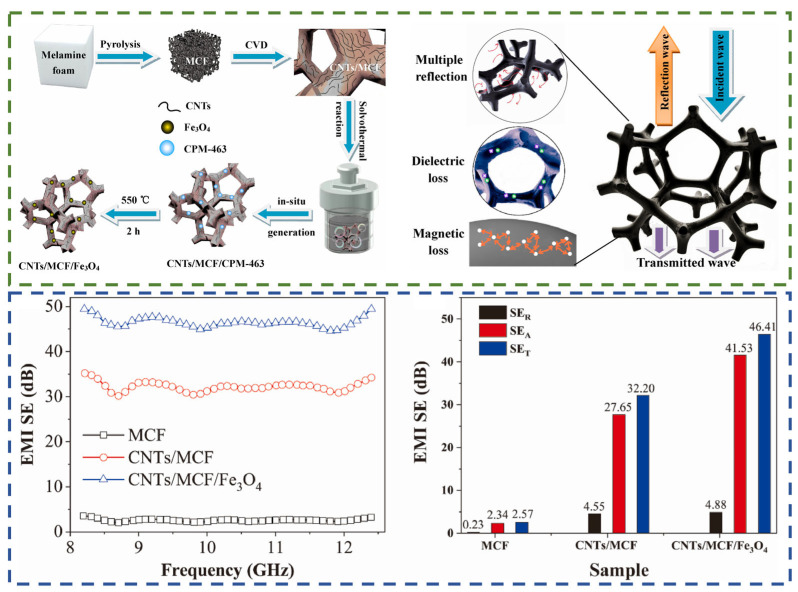
Schematic illustration of the CNT/MCF/Fe_3_O_4_ composite preparation method and its EMI shielding mechanism. The EMI shielding property values of MCF, CNTs/MCF, and CNT/MCF/Fe_3_O_4_ [117].

**Figure 12 nanomaterials-14-01291-f012:**
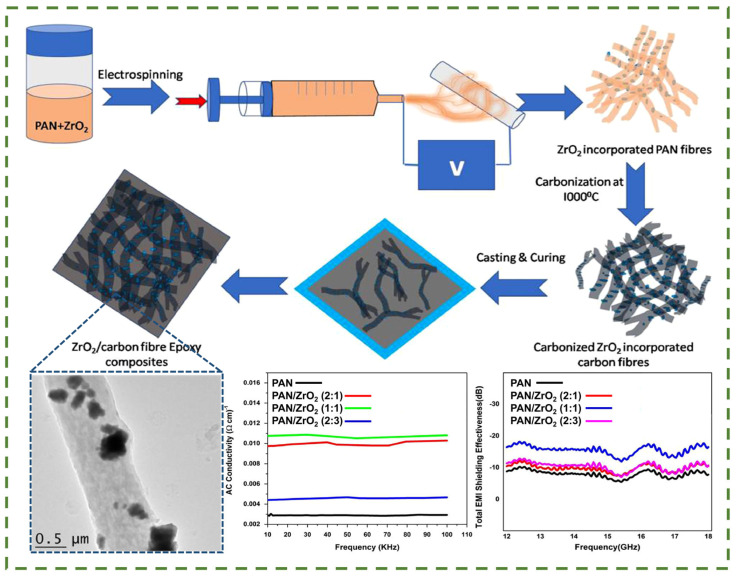
Schematic illustration of the method of obtaining the composite and its properties of electrical conductivity and electromagnetic protection [126].

**Figure 13 nanomaterials-14-01291-f013:**
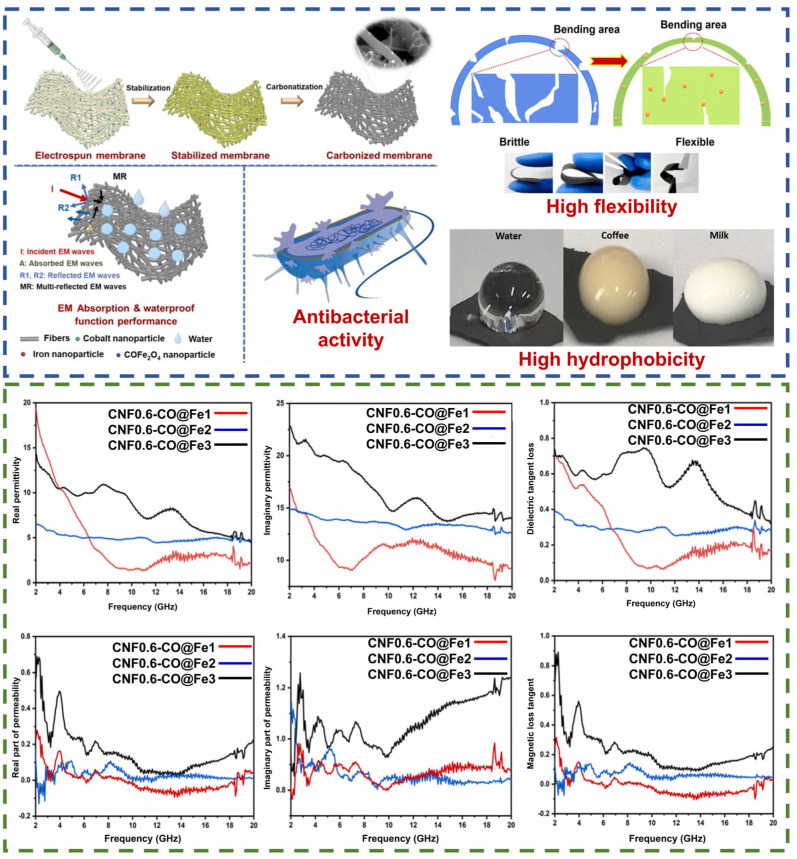
Schematic illustration of the method of composite production and its dielectric and magnetic loss tangent in the 2–20 GHz frequency range [128].

**Figure 14 nanomaterials-14-01291-f014:**
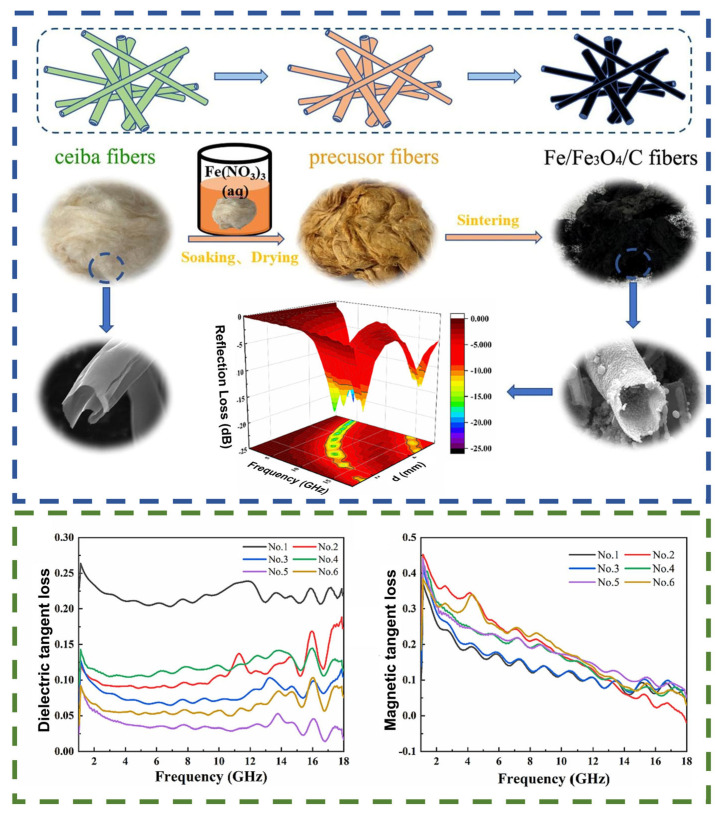
Schematic illustration of the method of composite production and its dielectric and magnetic loss tangent in the 2–18 GHz frequency range [129].

**Figure 15 nanomaterials-14-01291-f015:**
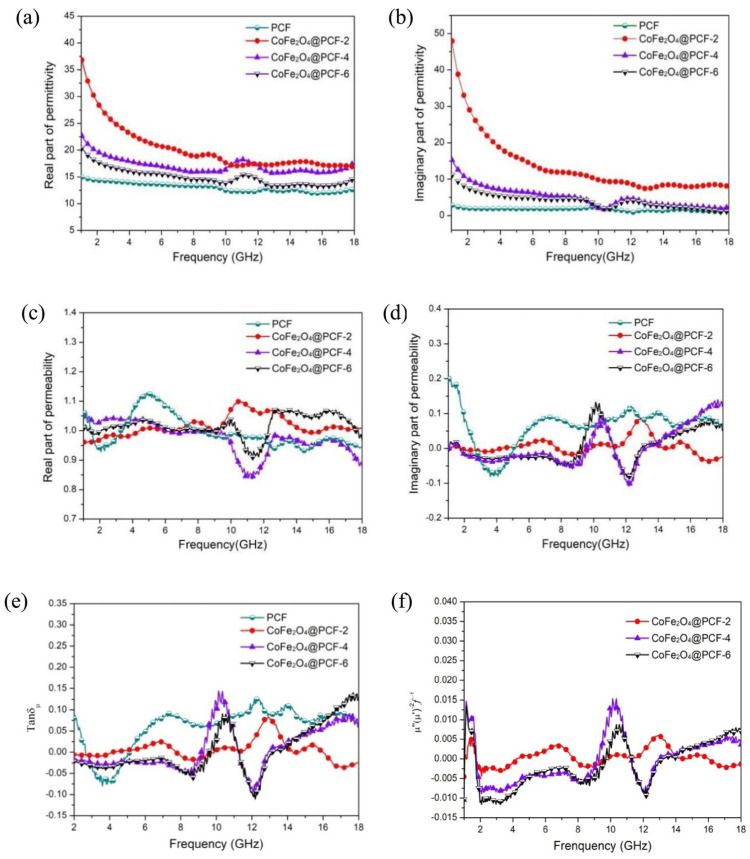
Frequency dependence of ε′ (**a**), ε″ (**b**), μ′ (**c**), μ″ (**d**), and tanδμ (**e**) of PCF and CoFe_2_O_4_@PCF nanocomposites, and C_0_ of CoFe_2_O_4_@PCF nanocomposites (**f**) [130].

**Table 1 nanomaterials-14-01291-t001:** A summary of the key electromagnetic properties of hybrid structures based on magnetite and carbon nanotubes, obtained by different methods.

Composites	Method of Synthesis	Electromagnetic Properties		Electrical Conductivity (S × m^−1^)	Saturation Magnetization (emu × g^−1^)	Comments	Ref.
		EMI SE (dB)	Bandwidth (GHz)				
MWCNTs/Fe_3_O_4_	Molecular and atomic fluorination followed by co-fluorochemical reaction	−38.7	10.5	0.183	77.2–75.7	Defects on the inner and outer tubes of MWCNTs increase polarization and conduction losses. The method allows for precise tuning of the location of defects on the nanotubes, which improves impedance matching and overall microwave absorption efficiency. This composite shows a minimum reflection loss (RL_min_) of −38.7 dB at 10.5 GHz when the thickness is 2.2 mm, and the maximum effective absorption bandwidth (EAB_max_) is 4.3 GHz at the thickness of 1.53 mm.	[110]
CNT/Necklace-like Fe_3_O_4_	One-pot hydrothermal method	−59.2	12.64	N/A	N/A	Enhanced dielectric properties and high electrical conductivity due to the CNTs. Good cycling stability and low resistance make it suitable for supercapacitors. This composite shows a minimum reflection loss (RL_min_) of −59.2 dB at 12.64 GHz with a thickness of 1.68 mm.	[111]
CNT/Fe_3_O_4_-Nanoflower	Solvothermal process	−58.6	15.28	N/A	N/A	The superior microwave absorbing properties are attributed to the coordination of CNTs and Fe_3_O_4_ nanoflower, abundant interfaces promoting interfacial polarization, and dielectric loss regulation. The minimal RL is −58.6 dB at 15.28 GHz. Meanwhile, the thickness is only 1.52 mm. Furthermore, when RL is below −10 dB, the actual absorption bandwidth is as high as 15 GHz at absorber thickness below 5 mm.	[112]
MWCNT@ Fe_2_O_3_/Fe_3_O_4_	Three-step synthesis	−72.17	8.80	N/A	23	Decoration of MWCNTs with Fe_2_O_3_/Fe_3_O_4_ nanoparticles significantly enhances dielectric and magnetic losses, resulting in improved microwave absorption performance. The strongest reflection loss and the broadest bandwidth reached −72.17 dB and 8.80 GHz for 2.8 mm and 2.4 mm matching thicknesses, respectively.	[113]
MWCNT/Fe_3_O_4_@ZnO	Chemoselective method for the synthesis of heterotrimers	−40.9	9.8	83.06	55.05	The size of Fe_3_O_4_ nanoparticles varies from 8.72 to 18.94 nm, and the ZnO layer has a thickness of about 3 nm.	[114]
MWCNT/Fe_3_O_4_	Solvothermal method	60.7	8–12	33.4	20.37	Homogeneous surface decoration, enhanced dielectric, magnetic, and AC conductivity performance. The highest EMI SE (60.7 dB) (Freestanding powder pellet with a low thickness of 500 μm) in the X-band region	[115]
Fe_3_O_4_/CNT@ Fe_3_O_4_/EP	Surface molecular engineering and mixing	−52.57	12.08–17.28	0.105	64.69	Enhanced impedance matching, high attenuation constant, improved thermal stability, and mechanical properties. S-Fe_3_O_4_ acts as a lubricant, enhancing dispersion. The composite achieves RL_min_ of −52.57 dB at 3.3 mm thickness and maximum EAB of 5.2 GHz at only 1.4 mm thickness.	[116]
CNTs/Fe_3_O_4_/Melamine-based carbon foam	In situ growth	46.41	8–12	83.06	N/A	High shielding effectiveness after 50 compression cycles and good mechanical robustness improved dielectric and magnetic loss capabilities. The total shielding effectiveness (SE_T_) of a functional material with a thickness of 3 mm from 32.20 dB to 46.41 dB in the X band (8.2–12.4 GHz).	[117]
3D CNT/Fe_3_O_4_	One-pot hydrothermal method	−56.8	11.12	N/A	N/A	Hierarchical urchin-like structure enhances EMW absorption. Optimal performance with 5 wt% CNTs. The formed Fe_3_O_4_/CNTs architecture was a robust EMW absorber with an RL of −56.8 dB at 11.12 GHz and a thin thickness of 2.15 mm.	[118]
CNTs-loaded Fe_3_O_4_	Chemical Vapor Deposition	−35.9	7.12	N/A	N/A	The composites demonstrated improved impedance matching, interface scattering, dielectric loss, and magnetic loss, contributing to enhanced electromagnetic wave absorption properties. This composite shows a minimum reflection loss (RL_min_) of −35.9 dB at 7.12 GHz with a thickness of 3 mm. The effective bandwidth of less than −10 dB is 4.32 GHz with a thickness of 1.5 mm.	[119]
CNTs/Fe@Fe_3_O_4_	Thermal decomposition method	−33	5.4	N/A	108	The pre-treatment with H_2_O_2_ introduces polar groups to HCNTs, enhancing dipole polarization. Core–shell Fe@Fe_3_O_4_ nanoparticles improve magnetic loss ability. Multiple dielectric and magnetic loss forms enhance electromagnetic absorption. This composite shows a minimum reflection loss (RL_min_) of −33 dB at 12.8 GHz with a thickness of 1.5 mm.	[120]
MWCNT/Fe_3_O_4_/TiO_2_	Sol–gel method/electrospinning	−8.2	10	N/A	N/A	The inclusion of TiO_2_ in the Fe_3_O_4_/MWCNT hybrid composite significantly enhances the material’s dielectric properties, promoting better phase transformation in PVDF and resulting in improved piezoelectric sensitivity and electromagnetic wave absorption. This composite shows a maximum reflection loss of −8.2 dB in the X band (8.2–12.4 GHz) with a thickness of 0.4 mm.	[121]
MWCNTs/Fe_3_O_4_/PVDF/PS/HDPE	Melt blending	25	9.5	0.01	N/A	The core–shell morphology between HDPE and PS was well retained after the addition of MWCNTs and Fe_3_O_4_. The absorption shielding was the main contributor to EMI SE improvement, resulting from dipole polarizations and interfacial polarizations caused by Fe_3_O_4_. This composite shows a maximum shielding effectiveness of 25 dB at 9.5 GHz with a thickness of 2.7 mm.	[122]

N/A—not available.

**Table 2 nanomaterials-14-01291-t002:** Comparative results on the effect of different fillers on the conductivity and shielding performance of carbon fiber composites.

Composites	Method of Synthesis	Electromagnetic Properties		Electrical Conductivity (S × m^−1^)	Saturation Magnetization (emu × g^−1^)	Comments	Ref.
		EMI SE (dB)	Bandwidth (GHz)				
CF/Fe_3_O_4_/GO	Electrohydrodynamic atomization deposition	32.9	10.7	N/A	N/A	The robust composite material comprising CFf/GO/Fe_3_O_4_/epoxy resin characterized by a notable tensile strength of 565.1 MPa and distinguished by its superior electromagnetic interference shielding properties, holds significant promise for application across the domains of aerospace engineering and telecommunications. The dimensions of the EMI specimens are 22.9 mm × 10.2 mm × 0.6 mm.	[131]
3D CF/nano-Fe_3_O_4_	Solvothermal synthesis	−62.6	8.2–12.4	N/A	39.7	The combination of high magnetic properties and effective impedance matching yielded remarkable electromagnetic shielding performance. The hybrid structure of 3D carbon nanofiber mats and Fe_3_O_4_ exhibited strong absorption capabilities, low density, and a broad absorption spectrum, suggesting significant potential for application in electromagnetic shielding. The optimal thickness is 2.5 mm, which provides the best microwave absorption performance with the lowest reflectivity of −47 dB at 10.0 GHz.	[136]
rCF/Fe_3_O_4_/acrylonitrile butadiene styrene	Ultrasonic exposure	37.9	8–12	N/A	58.76	A detailed study of the electrodynamic parameters revealed the attractive properties of reclaimed carbon fiber as a component of shielding materials, especially with the simultaneous addition of Fe_3_O_4_. The composite’s permittivity imaginary part increase was up to − 23.96 (88%) at 8 GHz compared to the virgin carbon fiber and Fe_3_O_4_ composite. The composites exhibit excellent shielding factors ranging from 33.7 to 37.9 dB at frequencies 8–12 GHz with an optimal thickness of 1 mm.	[137]
CFs@ MWCNTs/Fe_3_O_4_	The electrophoretic co-deposition process	33	8	N/A	8.74	The results revealed that the Fe_3_O_4_/multi-wall carbon nanotubes@carbon fiber nanocomposite structure with a high specific surface area of 87.12 m^2^ × g^−1^ was successfully fabricated. The composites exhibit shielding factors with a maximum SE of 35 dB at a thickness of 5 mm.	[138]
CFs@nano-Fe_3_O_4_@ PANI	The multi-step electrophoretic deposition	−11.11	~6	~7	0.191	The saturated magnetization (Ms) of the as-synthesized nano-Fe_3_O_4_ powder decreased from 72.612 emu × g^−1^ to 8.934 emu × g^−1^ for the nano-Fe_3_O_4_@CFs from 8.934 emu × g^−1^ to 0.191 emu × g^−1^for the PANI@nano-Fe_3_O_4_@CFs mats due to the reduction in the effective mass/volume percentage of nano-Fe_3_O_4_ particles in the composites. The composites exhibit an EMI SE of 29 dB at a thickness of 3 mm in the frequency range of 8.2–18 GHz.	[139]
CF/cement/nano-Fe_3_O_4_	Mechanical mixing and casting	29.8	8.2–12.4	N/A	N/A	With 0.4 wt% CF and 5 wt% Fe_3_O_4_ nanoparticles, the SE of the CF/Fe_3_O_4_/cement composite reached 29.8 dB at the frequency range of 8.2–12.4 GHz, which had a 34.4% increase than the CF/cement composite. The excellent EMI shielding property was attributed to the synergistic effect between CF and Fe_3_O_4_ nanoparticles. The Fe_3_O_4_/CF/cement composite, with a thickness of 7 mm, is believed to be a promising material for high EMI shielding.	[140]
CF/MXene/PANI	Dip-coating method	26.0	8.2–12.4	24.57	N/A	The results indicated that the 0.55 mm thickness flexible PANI/MXene/CF fabric possessed a good electrical conductivity (24.57 S × m^−1^), high EM SE (26.0 dB), favorable specific EMI SE (135.5 dB × cm^3^ × g^−1^) and excellent voltage-driven Joule heating properties.	[141]
CF/WS_2_	One-step hydrothermal method	36.0	2	N/A	N/A	The optimized EMI shielding performance is mainly derived from the enhanced electromagnetic wave absorption, which could be attributed to the loss of electromagnetic waves as caused by the rough surface morphology of CF and the unique wavy structure of WS_2_, the heterogeneous interface, multiphase structure and a large number of defects of WS_2_, as well as the good electrical conductivity of CF. The composite exhibits an EMI SE of 36.0 dB at a thickness of 3.00 mm in the frequency range of 2 GHz.	[142]
CF/PEKK via Mxene	The interfacial modification via hot-press procedures	65.2	8.2–12.4	0.13	N/A	The composites based on the CF and MXene nanosheets showed excellent flexural strength (1127 MPa), flexural modulus (81 GPa), and ILSS (89 MPa). Such great mechanical properties might be ascribed to the layer of MXene nanosheets, which could introduce mechanical interlocking, hydrogen bonds, and Van der Waals forces into the interface of MXene-modified CF. The volume fraction of CF was approximately 62.8% in the final composites of about 2 mm thickness.	[143]
CF/CNT	The method of selective growth of cluster arrays of CNTs on the CF surface	20	4.4	N/A	N/A	Surface functional groups of CF are used to control the density of CNT cluster arrays based on the route of metal–organic framework-derived Co-doped CNT. Due to the large numbers of CNT cluster arrays covering on the CF surface, a local conductive network is formed, which can effectively improve attenuation capability to electromagnetic waves. The composites exhibit excellent electromagnetic wave absorbing performance, with an effective absorption bandwidth of 4.4 GHz with a matching thickness of 1.38 mm.	[144]

N/A—not available.

## Data Availability

Data is contained within the article.

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
