# Peer review of "Magnetite-Incorporated 1D Carbon Nanostructure Hybrids for Electromagnetic Interference Shielding"

_nanomaterials, 2024, doi:10.3390/nano14151291_

Round 1

Reviewer 1 Report

Comments and Suggestions for Authors

It is a well written review paper. The reading of it is smooth. The  subject is important due to elaboration of electromagnetic shielding materials. The scientists who work on the materials can find a good source of up-to-date literature.

Author Response

Dear Reviewer,

Thank you for your positive feedback on our manuscript. We are pleased that you found it well-written and smooth to read. We appreciate your recognition of the importance of electromagnetic shielding materials and our effort to provide an up-to-date review.

We are glad that you consider our review a valuable resource for scientists in the field. Our goal was to compile the latest research to serve as a comprehensive reference. We remain open to any further suggestions for improvement.

Sincerely,

Bayan K.

Reviewer 2 Report

Comments and Suggestions for Authors

The present work explored the development and potential of magnetite-incorporated one-dimensional (1D) carbon nanostructure hybrids, focusing on their unique properties and synthesis methods. Various synthesis techniques, including solvothermal synthesis, in-situ growth, and electrostatic self-assembly, were discussed in detail, highlighting their impact on the structure and properties of the resulting composites. Overall, this review has certain reference function. However, some issues should be addressed.

1, The introduction of this review fails to articulate the significance and innovation behind studying magnetite-incorporated one-dimensional (1D) carbon nanostructure hybrids. What is the importance, novelty, and significance of researching these hybrids? Additionally, it is essential to highlight the advantages and disadvantages of magnetite and one-dimensional (1D) carbon nanostructures. Please rewrite the introduction section to emphasize the corresponding importance and innovation.

2, In the section on the Fundamental Principles of Electromagnetic Shielding, the author explains the principles of electromagnetic shielding, focusing on reflection, interference, and absorption. However, there is a lack of detailed explanation regarding the principles of electromagnetic wave absorption. It is recommended to include additional information on the principles of electromagnetic wave absorption and cite relevant literature to support these explanations, such as ACS Applied Materials & Interfaces, 2017, 9, 16404; Nanomaterials, 2024, 14, 587.

3, In the present work, 1D carbon nanostructures mainly include CNT and carbon fiber. Other 1D carbon nanostructures such as One-dimensional Carbon Nanowires, Carbon Nanorods and others are missing. Please explain the reason.

4, It was suggested to add the compositions in the comparison tables since the compositions have a great influence on the structure and properties of materials.

5, Among these materials being compared in Table 1 and Table 2, which one exhibits the best electromagnetic shielding performance, and what are the underlying principles absorption, reflection, or interference? What are the thickness and weight of the optimal material? Including this information in a comparative table would greatly benefit practical applications.

6, This review is lacking a crucial section - applications. Currently, these materials are only at the laboratory stage, and there is still a long way to go before they can be applied. The author can provide appropriate recommendations on the application of materials in the section addressing future challenges, these are more important than other issues.

Author Response

Dear Reviewer,

Thank you for your thorough and insightful review of our manuscript. We greatly appreciate your valuable comments and suggestions, which have significantly contributed to enhancing the quality and comprehensiveness of our work. 

We believe that these revisions have strengthened our manuscript and made it more relevant to both the academic community and industry practitioners. Your constructive feedback has been invaluable in this process, and we are grateful for your careful and detailed review.

Thank you once again for your contributions to improving our manuscript.

Sincerely,

Bayan K.

Comment #1: "The introduction of this review fails to articulate the significance and innovation behind studying magnetite incorporated one-dimensional (1D) carbon nanostructure hybrids. What is the importance, novelty, and significance of researching these hybrids? Additionally, it is essential to highlight the advantages and disadvantages of magnetite and one-dimensional (1D) carbon nanostructures. Please rewrite the introduction section to emphasize the corresponding importance and innovation."

Response #1: Thank you for your comment. We partially agree with the remark. The importance of studying magnetite-incorporated one-dimensional (1D) carbon nanostructure hybrids is primarily due to the need for the development of new shielding materials for EMI protection. We have added information about the novelty of researching these hybrids to the introduction section. The authors believe that covering all the advantages and disadvantages of magnetite and one-dimensional (1D) carbon nanostructures in the introduction would be challenging – these aspects are discussed in the main body of the article, as they are a result of considering the fundamental processes involved in the synthesis and study of the hybrids.

Line numbers: 79 - 87

"The novelty of research and application of hybrid composites based on one-dimensional carbon structures and nanomagnetite can be attributed to the combination and synergistic effect of the electrically conductive and magnetic properties of the components, which when combined result in an enhanced effect for protection against electromagnetic radiation. Furthermore, the combination of these properties allows for the formation of materials with universal shapes and the ability to function in a wide temperature range. The combination of these advantages, along with the variability of synthesis methods and their modifications, is of interest to researchers and reveals broad prospects for the creation of new hybrid composites."

Comment #2: "In the section on the Fundamental Principles of Electromagnetic Shielding, the author explains the principles of electromagnetic shielding, focusing on reflection, interference, and absorption. However, there is a lack of detailed explanation regarding the principles of electromagnetic wave absorption. It is recommended to include additional information on the principles of electromagnetic wave absorption and cite relevant literature to support these explanations, such as ACS Applied Materials & Interfaces, 2017, 9, 16404; Nanomaterials, 2024, 14, 587."

Response #2: The authors agree with the remark and have added relevant references as recommended by the reviewer. This has allowed for a more in-depth discussion and better illumination of the principles of electromagnetic wave absorption.

Line numbers: 145 - 161

As stated by the authors of the study (https://doi.org/10.1021/acsami.7b02597), dielectric losses play a pivotal role in the process of electromagnetic wave absorption. The underlying mechanism of this process is absorption, which is dependent on the magnetic and dielectric losses within the material, as well as the direction of domains, walls, and resonance phenomena. In order to estimate these losses, it is necessary to determine the amount of electromagnetic waves incident on the absorber, which is reflected in the impedance matching coefficient. The incident electromagnetic wave penetrates the absorber, where it may either be scattered or converted to heat. By determining the impedance matching coefficient, the efficiency of electromagnetic wave absorption can be enhanced.

Additionally, the principles of electromagnetic absorption can be influenced by the structure of the material, particularly porous structures, due to their high specific surface area and porous fiber composition. Electromagnetic waves undergo repeated scattering and reflection, ultimately converting into heat energy. This process contributes to more efficient absorption of electromagnetic waves. For example, the principles of electromagnetic absorption have been investigated using carbonized chitosan fibers obtained by electrospinning (https://doi.org/10.3390/nano14070587) as an illustrative example. This porous structure promotes the absorption of eddy currents, leading to an improved impedance matching coefficient.

Comment #3:  "In the present work, 1D carbon nanostructures mainly include CNT and carbon fiber. Other 1D carbon nanostructures such as One-dimensional Carbon Nanowires, Carbon Nanorods and others are missing. Please explain the reason."

Response #3: Thank you for your comment. The focus of our review on carbon nanotubes (CNT) and carbon fibers (CF) was due to their predominant presence in the literature and their well-documented performance in electromagnetic interference (EMI) shielding applications. These materials have been extensively studied and have demonstrated significant potential and versatility in this field.

However, we acknowledge the importance of other one-dimensional (1D) carbon nanostructures such as carbon nanowires and carbon nanorods. These materials indeed possess unique properties and potential for various applications, including EMI shielding. The decision to concentrate on CNTs and CFs was made to provide a detailed and comprehensive review within the scope and length limitations of the manuscript.

Line numbers: 690 - 757
The authors appreciate your comment and, in response, have decided to enhance the quality of the article by expanding Section 5. We have added Subsections 5.3 and 5.4, which are dedicated to composites with carbon nanorods and nanowires. 

Comment #4: "It was suggested to add the compositions in the comparison tables since the compositions have a great influence on the structure and properties of materials."

Response #4: Thank you for your valuable suggestion to include compositions in the comparison tables. We agree that the compositions have a significant influence on the structure and properties of materials, and including this information will enhance the clarity and usefulness of the tables.

Comment #5: Among these materials being compared in Table 1 and Table 2, which one exhibits the best electromagnetic shielding performance, and what are the underlying principles – absorption, reflection, or interference? What are the thickness and weight of the optimal material? Including this information in a comparative table would greatly benefit practical applications.

Response #5: Thank you for your valuable comment. We have taken your suggestions into account and updated the Comments section in Table 1 and Table 2 with information related to the thickness of the resulting composites and their effectiveness. This additional information provides a more comprehensive understanding of the optimal materials for electromagnetic shielding applications.

Comment #6: This review is lacking a crucial section - applications. Currently, these materials are only at the laboratory stage, and there is still a long way to go before they can be applied. The author can provide appropriate recommendations on the application of materials in the section addressing future challenges, these are more important than other issues.

Response #6: Thank you for your valuable comment. We agree that discussing the potential applications of these materials is crucial for providing a comprehensive understanding of their future impact. In response, we have integrated a detailed discussion of the applications and future challenges of these materials into the "Conclusion and Prospects" section of the manuscript (Lines 848-880). We believe these additions enhance the manuscript by providing a thorough outlook on the future applications and necessary advancements for the practical implementation of these composites.

Reviewer 3 Report

Comments and Suggestions for Authors

This is a nice review about iron oxide incorporated one-dimensional carbon tube nanostructures for applications in electromagnetic interfering shielding. The basic concepts and constructing materials were described clearly. The preparation methods were introduced in detail. The challenges are also addressed well. This review is helpful for junior researchers in this field. The manuscript is organized well. The text falls in the scope of this Journal. Thus, I’d like to suggest acceptance of this manuscript for publication in Nanomaterials in the present form.   

Author Response

Dear Reviewer,

Thank you for your thorough review and positive feedback on our manuscript. We are delighted that you found our review on iron oxide incorporated one-dimensional carbon tube nanostructures for electromagnetic interference shielding both informative and well-organized.

We appreciate your acknowledgment of the clear description of basic concepts, constructing materials, and detailed preparation methods. It is gratifying to hear that the challenges addressed are useful for junior researchers in the field.

We are pleased that you find the manuscript falls within the scope of Nanomaterials and support its acceptance for publication in its present form.

Thank you once again for your encouraging comments.

Sincerely,

Bayan K.

Round 2

Reviewer 2 Report

Comments and Suggestions for Authors

All issues were well addressed, and this work can be accepted.